

# Early-diverging plesiosaurs from the Pliensbachian (Lower Jurassic) of northwestern Germany

Sven Sachs[1], Jahn J. Hornung[2] and Daniel Madzia[3]

[1] Abteilung Geowissenschaften, Naturkunde-Museum Bielefeld, Bielefeld, Germany
[2] Niedersächsisches Landesmuseum Hannover, Hannover, Germany
[3] Department of Evolutionary Paleobiology, Institute of Paleobiology, Polish Academy of Sciences, Warsaw, Poland

## ABSTRACT

The knowledge of Pliensbachian (Early Jurassic, ∼192.9–184.2 Ma) plesiosaurs is notoriously insufficient. Although there have been specimens described from different parts of the world, only three of them have been established as diagnosable taxa. Here, we describe two previously unreported lower Pliensbachian plesiosaur occurrences that originate from two sites located in North Rhine-Westphalia, Germany. One of the new occurrences is represented by three cervical and three indeterminable vertebrae from Werther, the other includes two associated pectoral or anterior dorsal vertebrae from Bielefeld. Although highly incomplete, the Werther individual, which derived from the *Uptonia jamesoni* Zone, is found to represent the only reliably identified early Pliensbachian pliosaurid known to date. Its material is geographically and stratigraphically proximate to the late Pliensbachian pliosaurid *Arminisaurus schuberti*, found in a clay-pit located in the Bielefeld district of Jöllenbeck. However, even though the Werther plesiosaur and *A. schuberti* show a broadly similar morphology of the preserved cervical section, a precise identification of the Werther taxon is currently impossible.

## INTRODUCTION

Plesiosaurs were a diverse clade of aquatic tetrapods whose fossil record spans from the Upper Triassic to the Cretaceous/Paleogene boundary (*e.g.,* *Ketchum & Benson, 2010*; *Benson & Druckenmiller, 2014*; *Madzia & Cau, 2020*). Yet, their stratigraphic distribution is uneven. For instance, with respect to their Lower Jurassic record, rich plesiosaur material is known from Hettangian and Toarcian strata. Notoriously poor, however, is their record dated to the Pliensbachian (∼192.9–184.2 Ma). So far, only three plesiosaur taxa have been established from this stage: *Westphaliasaurus simonsensii* from the lower Pliensbachian of Sommersell in western Germany (*Schwermann & Sander, 2011*), *Cryonectes neustriacus* from the upper Pliensbachian of Fresney-le-Puceux in northern France (*Vincent, Bardet & Mattioli, 2013*), and *Arminisaurus schuberti* from the upper Pliensbachian of Bielefeld in northwestern Germany (*Sachs & Kear, 2018*).

Corresponding author
Sven Sachs, sachs.pal@gmail.com

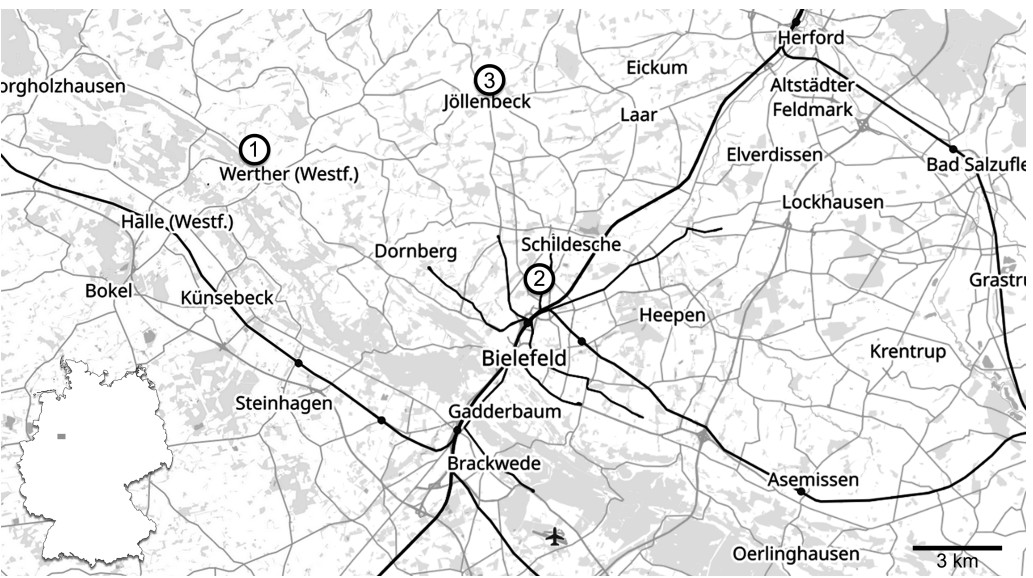

**Figure 1** **Locality map.** Map showing the localities at Werther (1) and Bielefeld-Sudbrack (2), as well as the type locality of *Arminisaurus schuberti* in Bielefeld-Jöllenbeck (3) with their position within the map of Germany. Map by OpenStreetMap (CC BY-SA 2.0).

In addition to these taxa there are a number of incomplete and fragmentary specimens known from Germany (*e.g.*, *Janensch, 1928*; *Schubert, 2007*), Spain (*Schulz, 1858*; *Bardet et al., 2008*), England (*Storrs, 1995*; *Forrest, 2006*; *Evans, 2012*), Denmark (*Rees & Bonde, 1999*; *Smith, 2008*), Greenland (*Bendix-Almgreen, 1976*), and Australia (*Thulborn & Warren, 1980*; *Kear, 2012*).

Here, we describe and illustrate previously unreported historical plesiosaur specimens from the lower Pliensbachian of northwestern Germany (Figs. 1–3). The material originates from two fossil sites and includes three cervical vertebrae and additional indeterminate vertebrae found at Werther (GZG.V.000092), and two associated post-cervical vertebral centra, either pectorals or anterior dorsals, found in the vicinity of Bielefeld (Namu ES/jL-3868).

## Geological and stratigraphic setting

The studied specimens (GZG.V.000092 and Namu ES/jL-3868) originate from successions preserved within the Herford Syncline (Herforder Liasmulde, *e.g.*, *Schubert, 2007*) and its southwestern boundary, the Osning Fault Zone, a multi-phasic normal/reverse/strike-slip fault system (Fig. 2). Both structural elements attained their current configuration during the Late Cretaceous and underwent exhumation since the late Paleogene (*e.g.*, *Drozdzewski & Dölling, 2018*). During the Early Jurassic the area was part of the Central European Archipelago, an array of islands separated by marine straits and local basins (*e.g.*, *Ziegler, 1982*). Within the area of the Herford Syncline and its surroundings, marine sedimentation during the Sinemurian and Pliensbachian mostly comprised claystones and marlstones. The discontinuity surface at the base of the Pliensbachian is a notable exception from this rather

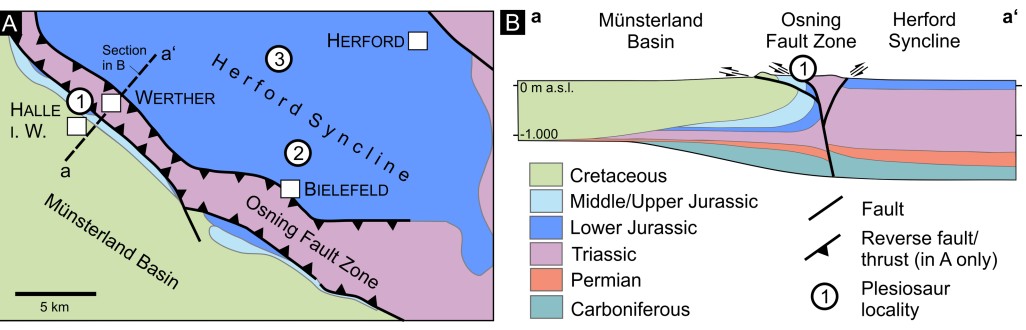

**Figure 2  Regional geological situation.** (A) Geological sketch-map of the Herford Syncline and the Bielefeld section of the Osning Fault Zone with the plesiosaur fossil sites discussed in this paper, indicated by numbers (1 to 3, corresponding to Fig. 1). Based on *Drozdzewski & Dölling (2018)* and *Geoportal NRW (2023)*. (B) Geological cross-section of the Osning Fault Zone at the line a-a' (Fig. 1A), showing the approximate position of the former Spilker clay-pit (locality 1), slightly projected to the SE. Based on *Drozdzewski & Dölling (2018)*.

monotonous succession. Caused by a local regression, it is overlain by a highly condensed section (Rottorf Formation, *Mönnig, 2023a*), representing the lowermost Pliensbachian (*Uptonia jamesoni* Zone). These marginal facies passed basin-wards and upwards into the claystones and marlstones of the Capricornumergel Formation, representing the continuous distal facies of the lower Pliensbachian (*U. jamesoni* to *Tragophylloceras ibex* to *Prodactylioceras davoei* Zones, *Mönnig, 2023b*). The fine-grained, clayey-marly lithofacies continues upward into the overlying Amaltheenton Formation (*Amaltheus margaritatus* Zone, upper Pliensbachian; Fig. 3).

The material from the former Spilker clay-pit at Werther lacks precise stratigraphic information. However, the lithology and fossil content of the matrix attached to the skeletal remains allow determining their stratigraphic position within the succession that was once exposed at this outcrop. According to *Büchner, Hoffmann & Jordan (1986)*, the clay-pit exposed the Osning Fault Zone, resulting in steep to nearly vertical dip of the beds. By reverse faulting, lower Middle Triassic sandstones (Röt Formation) in the NE were upfaulted against the Lower Jurassic succession in the SW. The latter comprised an upper Sinemurian section of unspecified thickness, overlain by 1.5 m of the condensed Rottorf Formation (lower Pliensbachian, *U. jamesoni* Zone), which is, in turn, overlain by 115 m of dark, partially pyritiferous claystones with calcareous concretions of the Capricornumergel Formation (lower Pliensbachian, *T. ibex* and *P. davoei* zones; *Meyer, 1907*; *Büchner, Hoffmann & Jordan, 1986*).

The Rottorf Formation consists of bioclastic clay- and marlstones with high amounts of dispersed Fe-hydroxide (''iron stones''), causing a distinct pink to reddish color on the weathered surface, and in many layers contain Fe-ooids or interbedded Fe-oolites. According to *Büchner, Hoffmann & Jordan (1986)*, there are also abundant calcareous pebbles that are often altered by bioerosion (borings) and wood fragments.

The plesiosaur remains from the Spilker clay-pit (GZG.V.000092) are associated with pinkish, bioclastic, particle-bearing claystones and well-rounded calcareous pebbles, with

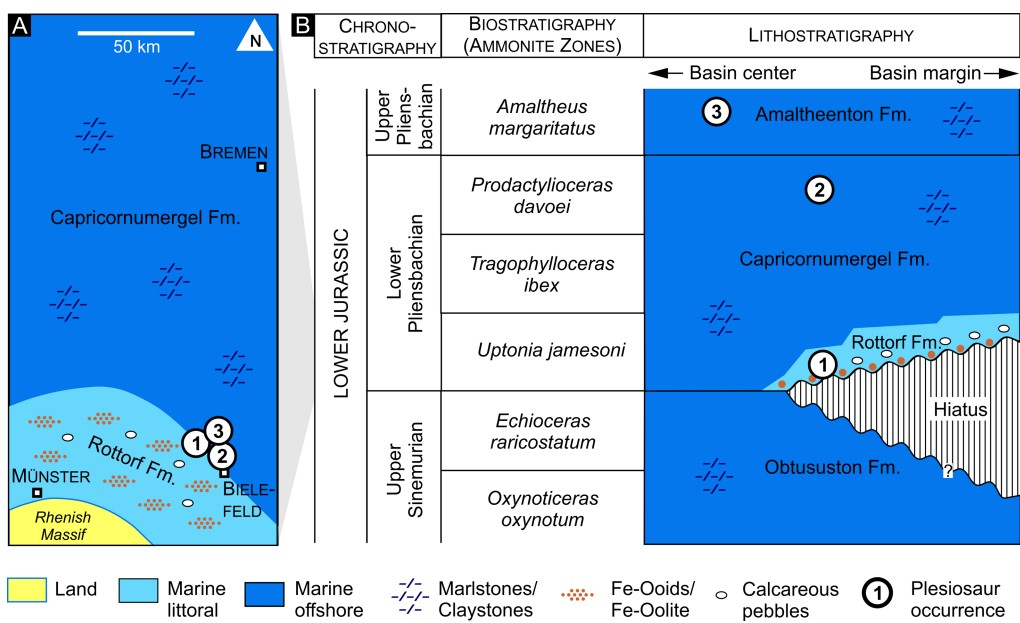

**Figure 3** **Geology and stratigraphy.** (A) Paleogeography and lithofacies of what is today northwestern Germany during the early Pliensbachian. Based on *Büchner, Hoffmann & Jordan (1986)*. (B) Chrono-, bio- and lithostratigraphy of the lower Pliensbachian in the Herford Syncline and adjacent regions after *Hoffmann (1982)*, *Deutsche Stratigraphische Kommission (2016)*, *Mönnig (2023a)*, and *Mönnig (2023b)*. Locations of the plesiosaur fossil sites discussed in this paper are indicated by numbers (1 to 3, corresponding to Fig. 1).

a Fe-hydroxide coating, adhering to the skeletal fragments. Accompanying invertebrate fossils include shell debris (identifiable are the bivalves *Oxytoma* sp. and *Eopecten*? sp., and a pleurotomariid gastropod), indeterminate belemnites, as well as the ammonites *Phricodoceras* cf. *taylori* (*Sowerby, 1829*), *Platypleuroceras* cf. *caprarium* (*Quenstedt, 1856*), and *Platypleuroceras* sp.; consequently, the plesiosaur material can stratigraphically be unambiguously referred to the Rottorf Formation. The accompanying ammonites indicate the *taylori* Subzone of the *jamesoni* Zone (*Hoffmann, 1982*). It is worth noting that *Büchner, Hoffmann & Jordan (1986)* have previously reported the *brevispina* Subzone as the lowermost section of the Rottorf Formation. If the identification of the *taylori* Subzone is correct, the 1.5 m thick Rottorf Formation at the Spilker clay-pit represents the condensed section of three subzones (*taylori*, *polymorphus*, and *brevispina*) of the *jamesoni* Zone.

In any case, the plesiosaur specimens from the *jamesoni* zone of the Spilker clay-pit represent the stratigraphically oldest of the specimens under consideration herein, followed by the material (Namu ES/jL-3868) from the Bielefeld-Sudbrack locality (*P. davoei* Zone, Capricornumergel Formation, lower Pliensbachian), and the type horizon of *Arminisaurus schuberti* (*Amaltheus subnodosus* Subzone, *Amaltheus margaritatus* Zone, Amaltheenton Formation, upper Pliensbachian; *Sachs & Kear, 2018*).

## METHODS

### Phylogenetic analyses

The Werther plesiosaur (GZG.V.000092) is geographically and stratigraphically proximate to the late Pliensbachian pliosaurid *Arminisaurus schuberti* from the Beukenhorst II clay-pit that is located in the Bielefeld district of Jöllenbeck, with which it also shares the majority of character states (see Discussion for detailed information). In order to assess whether these similarities have phylogenetic significance, we supplement our descriptions and comparisons through exploration of the placement of GZG.V.000092 among plesiosaurs using the dataset of *Sachs, Eggmaier & Madzia (2024)*, which represents a substantially modified version of the matrix originally assembled by *Benson & Druckenmiller (2014)*, and includes first-hand scores of *Arminisaurus schuberti* obtained from *Sachs et al. (2023)*. The final version of the matrix, which differs from that of *Sachs, Eggmaier & Madzia (2024)* only in the addition of the Werther plesiosaur, includes 131 operational taxonomic units (OTUs) and 270 characters; 67 of which were set as 'additive' (= 'ordered') following *Madzia, Sachs & Lindgren (2019)*.

Our analyses were performed using maximum parsimony as the optimality criterion and through TNT 1.6 (*Goloboff & Morales, 2023*). We have conducted four runs. The first run wase based on equal weights; the other three runs used the implied weighting function, with the concavity parameter ($K$) set to 6, 9, and 12. In all our analyses, we used *Neusticosaurus pusillus* as the outgroup. For each of the phylogenetic analyses, we fixed the maximum number of most parsimonious trees to 200,000 (command ''hold 200000''). Then, we ran the 'New Technology' (NT) search which involved 500 addition sequences and default settings for sectorial searches, ratchet, drift, and tree fusing (all activated). Following the NT search, we performed a 'Traditional Search' with tree bisection-reconnection (TBR) branch-swapping on trees saved to RAM. For the phylogenetic analysis using equal weights, nodal support was assessed through the Bremer support values (with TBR and retaining sub-optimal trees incorporating up to 3 additional steps). Nodal support for the parsimony analyses with implied weighting was assessed through Symmetric Resampling, using a 'Traditional Search', 1,000 replicates, a default change probability (set at 33), and output expressed as frequency differences (GC).

See Supplemental Informations 1 and 2 for the character list and a TNT-executable code, respectively.

### Systematic paleontology

Sauropterygia *Owen, 1860*
Plesiosauria *de Blainville, 1835*
Pliosauridae *Seeley, 1874*
Pliosauridae indet.

**Material.** GZG.V.000092, three cervical vertebrae and three indeterminable vertebrae (Figs. 4 and 5). The fragments are catalogued under a single number, owing to the

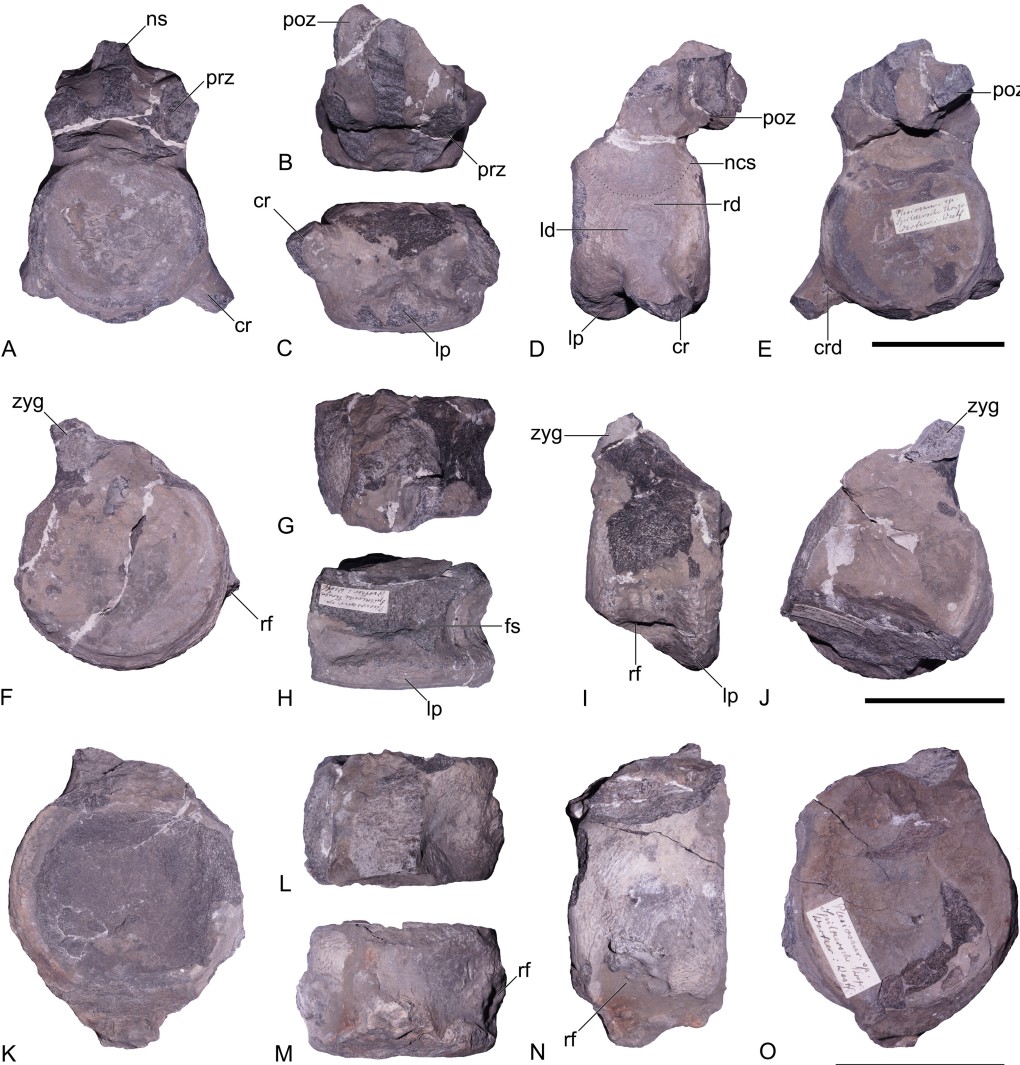

**Figure 4** **Pliosauridae indet., GZG.V.000092, cervical vertebrae; Rottdorf Formation (*Uptonia jamesoni* Zone, lowermost Pliensbachian) of former Spilker clay-pit, Werther, northwestern Germany.** (A–E) Anterior cervical vertebra (GZG.V.000092a) in (A) anterior, (B) dorsal, (C) ventral, (D) lateral, and (E) posterior view. (F–J) Supposed mid-cervical vertebra (GZG.V.000092b) in (F) anterior, (G) dorsal, (H) ventral, (I) lateral, and (J) posterior view. (K–O) Supposed posterior cervical vertebra (GZG.V.000092c) in (K) anterior, (L) dorsal, (M) ventral, (N) lateral, and (O) posterior view. Scale bars equal five cm. Abbreviations: cr, cervical rib; crd, posterior depression at cervical rib; fs, foramen subcentrale; ld, lateral depression; lp, lip-like projection; ncs, neurocentral suture; ns, neural spine; poz, postzygapophysis; prz, prezygapophysis; rd, ridge-like structure; rf, rib facet; zyg, zygapophysis.

assumption that they belong to a single individual. The separate fragments are distinguished by suffix letters 'a' through 'f', as indicated in Figs. 4 and 5.

**Locality and horizon.** Former Spilker clay-pit, Werther (Westfalen), Gütersloh district, North Rhine-Westphalia, Germany; Rottorf Formation, lower Pliensbachian (*Uptonia jamesoni* Zone), Lower Jurassic.

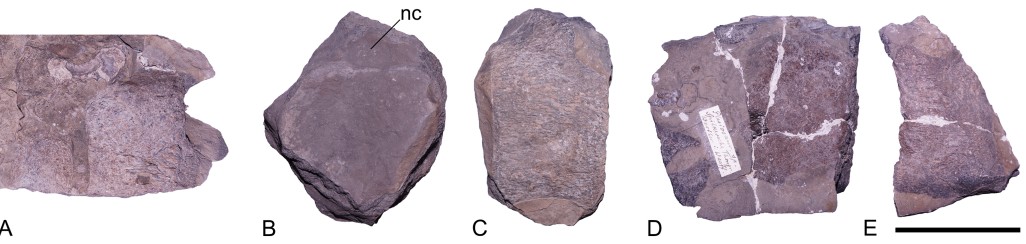

**Figure 5** **Pliosauridae indet., GZG.V.000092, indeterminate vertebrae; Rottdorf Formation (*Uptonia jamesoni* Zone, lowermost Pliensbachian) of former Spilker clay-pit, Werther, northwestern Germany.** (A) A damaged vertebra (GZG.V.000092d), (B–C) a centrum (GZG.V.000092e) in (B) articular and (C) damaged lateral view, (D–E) a damaged centrum (GZG.V.000092f) in (D) articular and (E) lateral view. Scale equals five cm. Abbreviation: nc, neural canal.

**Remarks.** GZG.V.000092 lacks detailed documentation, including information regarding its discovery.

## Description and comparisons

Six incomplete vertebrae are preserved; three of them can be identified as cervicals, shown by the lateroventrally placed rib facets (Fig. 4). The remaining three are not preserved well enough to be identified with certainty (Fig. 5). One of the cervicals (Figs. 4K–4O) is considerably larger and shows more prominent rib facets. Consequently, this vertebra likely derives from the posterior section of the neck. The size of the two other cervicals (Figs. 4A–4J) indicates that they likely originate from the anterior or middle section of the neck.

The articular faces of the centra are exposed in all three cervicals and in one of the indeterminable vertebrae (Fig. 5B). In the other specimens they are either damaged or obscured by matrix.

All cervical centra are wider and higher than long, a condition found in several early-diverging plesiosaurs, including *Rhaeticosaurus mertensi* and the pliosaurids *Arminisaurus schuberti* and *Cryonectes neustriacus* (*Vincent, Bardet & Mattioli, 2013*; *Wintrich et al., 2017*; *Sachs & Kear, 2018*).

As visible, the articular facets are amphicoelous and surrounded by a thickened rim. Anteroventrally, a prominent lip is present in two vertebrae (Figs. 4C, 4D and 4H). A similar prominent lip is found in the pliosaurid *Arminisaurus schuberti* (*Sachs & Kear, 2018*). The posterior articular face is preserved in one of the vertebrae (Fig. 5E) and here a ventral lip is absent.

The lateral sides of the cervical centra are anteroposteriorly concave. In the best preserved cervical (Fig. 4D), there is a circular lateral depression present dorsal to the rib facet on both sides. Dorsal to this concavity, there is a structure resembling a lateral ridge. However, neither of these structures is present in the other cervicals. Therefore, it is unclear if they are a taphonomic artefact. A similar circular depression can otherwise be found in some cervicals of *Cryonectes neustriacus* (*Vincent, Bardet & Mattioli, 2013*, fig. 9) and *Brancasaurus brancai* (S Sachs, pers. obs., 2013). In one of the cervicals, remnants of the cervical ribs are still fused to the centrum (Figs. 4A–4E). The rib facets in all cervicals are
placed lateroventrally and slightly more posterior to the mid-length of the centrum. In all cervicals, the morphology of the rib facets is somewhat obscured by either the cervical ribs or by matrix. However, a slight depression in the posterior margin of the cervical ribs indicates that two co-joined rib facets were formed (Fig. 4E). This condition is found in most pliosaurids and rhomaleosaurids, but also in some Early Jurassic plesiosauroids, such as microcleidids or *Westphaliasaurus simonsensii* (*Benson & Druckenmiller, 2014*: Appendix 2, character 160).

The dorsolaterally-placed neural arches are fused to the centra but a semicircular neurocentral suture is still indicated in the cervical vertebrae (Figs. 4A). A circular neural canal is visible in two cervicals (Figs. 4A and 4F) and one of the indeterminate vertebrae (Fig. 5B). The zygapophyses are hard to assess; they are either largely broken off or otherwise damaged. One right postzygapophysis is reasonably well preserved (Figs. 4B, 4D and 4E). It exceeds the level of the centrum posteriorly with most of its length. Laterally, the postzygapophyses are subequal to the width of the centrum having articular faces that are planar. All zygapophyses face dorsoventrally. The neural spines are broken off in all vertebrae, preserving only their base in one anterior cervical (Fig. 4B).

In ventral view, the anterior and posterior edges of the articular surface rims are transversely widened and form a triangular-shaped bulge. In the anterior articular face this bulge extends further ventrally and forms the lip-like protrusion described above. Both, the anterior and posterior articular surface bulges extend towards the mid-length of the centrum where they merge with a thickened and rounded midline keel (Figs. 4C and 4H). This condition is also present in the geographically and stratigraphically proximal Pliensbachian pliosaurid *Arminisaurus schuberti.* A similar rounded ventral midline keel is also present in the cervicals of later-diverging pliosaurids; *e.g.*, *Peloneustes philarchus* and *Eardasaurus powelli* (*Linder, 1913*; *Ketchum & Benson, 2022*). There are two subcentral foramina.

Plesiosauria indet.

**Material.** Namu ES/jL-3868, two associated pectoral or anterior dorsal vertebrae (Fig. 6)

**Locality and horizon.** Former Klarhorst clay-pit, Sudbrackgebiet, Bielefeld, North Rhine-Westphalia, Germany; Capricornumergel Formation, lower Pliensbachian (*Prodactylioceras davoei* Zone), Lower Jurassic.

**Remarks.** Namu ES/jL-3868 was found in the early 1930s in association with skeletal remains of a giant specimen of the ichthyosaur *Temnodontosaurus. Hungerbühler & Sachs (1996)* assigned them to the ichthyosaur specimen.

## Description and comparisons

Two associated centra with attached neural arch pedicles are preserved. The lateral apophyses are broken off, but they have been placed dorsally which indicates that these vertebrae are either pectoral vertebrae (*sensu Sachs, Kear & Everhart, 2013*) or anterior dorsal vertebrae (compare *e.g.*, *Smith & Benson, 2014*: pl. 13, Fig. 2). Both centra are wider than long and high and higher than long. The articular faces of the centra are mostly obscured but they appear only slightly amphicoelous (Fig. 6), being surrounded by

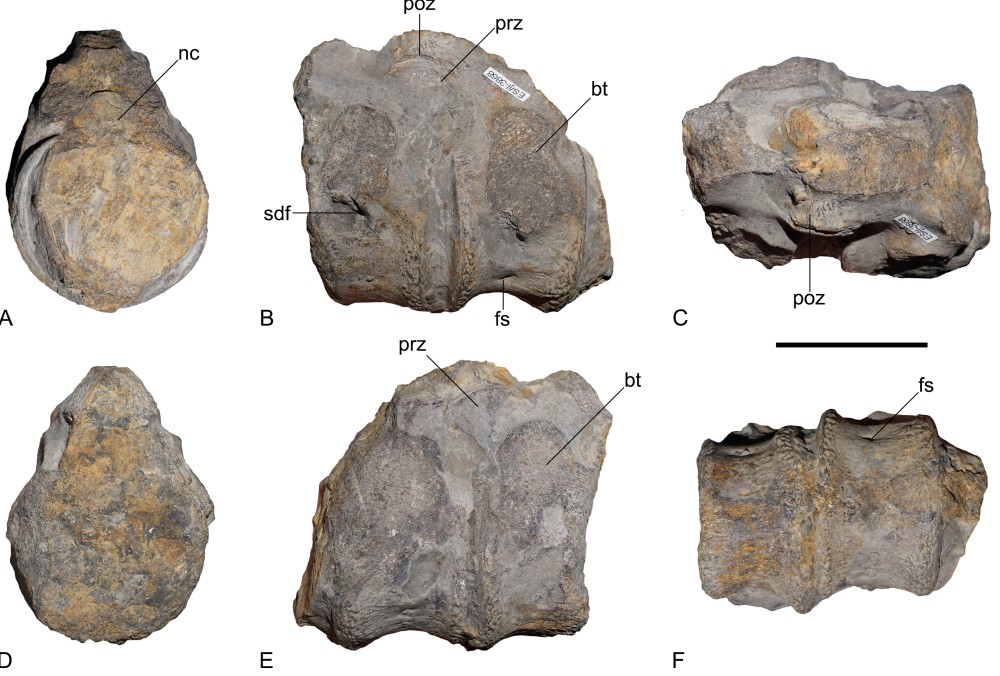

**Figure 6  Plesiosauria indet., Namu ES/jL-3868, pectoral or anterior dorsal vertebrae; Capricornu- mergel Formation (*Prodactylioceras davoei* Zone, upper lower Pliensbachian) of Bielefeld-Sudbrack, northwestern Germany.** (A) Anterior, (B) right lateral, (C) dorsal, (D) posterior, (E) left lateral, and (F) ventral view. Scale bar equals five cm. Abbreviations: bt, broken rib facet or transverse process; fs, foramen subcentrale; nc, neural canal; poz, postzygapophysis; prz, prezygapophysis; sdf, subdiapophyseal fossa.

flattened articular surface rims. Laterally, both centra are largely occupied by the broken rib facets/transverse processes, which are placed dorsally, adjacent to the neural canal. Transverse processes of dorsal vertebrae being placed adjacent to the neural canal is a common condition in plesiosaurs (*Benson & Druckenmiller, 2014*: Appendix 2, character 181). This differs, however, from the dorsal vertebrae of *Arminisaurus schuberti* where the transverse processes are placed dorsal to the neural canal (*Sachs & Kear, 2018*: Fig. 5A). Ventral to the remnants of the rib facets/transverse processes, a transverse buttress is indicated. This buttress resembles the condition observable in *R. thorntoni* (see *Smith & Benson, 2014*, pl. 13, fig. 2). Adjacent to the buttress, a subdiapophyseal fossa (*sensu Hampe, 2013*) is formed on each side (Fig. 6B). In *A. schuberti*, the ventral sides of the transverse processes of the dorsal vertebrae show a transverse buttress and an associated anterior fossa (*Sachs & Kear, 2018*).

A circular neural canal is visible in the anterior of the two vertebrae (Fig. 6A). The prezygapophyses are preserved in the posterior vertebra (Fig. 6). They are lobate in lateral view and extend over the level of the centrum with about half of their length. The postzygapophyses of the anterior-more vertebra articulate with the prezygapophyses of the second vertebrae and are thus only partly visible. They extend over the level of the centrum with most of their length. The postzygapophyses of the posterior-more vertebra are largely damaged. The articular faces are not visible in any of the zygopophyses. The

**Table 1  Numerical results of the parsimony analyses.**

| Run | MPT (NT) | BS | MPT (TS) | CI | RI |
|---|---|---|---|---|---|
| EW | 57 | 2088 | 200,000 | 0.191 | 0.684 |
| IW ($K = 6$) | 26 | 137.11275 | 113,967 | 0.190 | 0.681 |
| IW ($K = 9$) | 23 | 109.07487 | 32,319 | 0.190 | 0.682 |
| IW ($K = 12$) | 45 | 90.80995 | 4,617 | 0.191 | 0.683 |

Notes.
  BS, best score (tree length); CI, Consistency Index; EW, parsimony analysis using equal weighting; IW, parsimony analysis using implied weighting; MPT, number of most parsimonious trees; NT, 'New Technology' search; RI, Retention Index; TS, Traditional Search.

neural spines are broken off in both vertebrae. The ventrolateral and ventral sides of the centra are concave and bear foramina subcentralia (Fig. 6).

## Results of phylogenetic analyses

See Table 1 for the numerical results of our phylogenetic analyses, Fig. 7 for reduced tree topologies focusing on the pliosaurid segments of the trees that are relevant for the assessment of the phylogenetic placement of the Werther plesiosaur, and Supplemental Information 3 for full tree topologies and nodal support values.

The parsimony analysis using equal weights has produced a very poorly resolved strict consensus tree, failing to reconstruct a monophyletic Pliosauridae. However, inspection of the most parsimonious trees (MPTs) has shown that the basal topological instability was mainly caused by the nesting of *Thalassiodracon hawkinsii* that was alternatively inferred as either, an early-diverging pliosaurid or an early-diverging plesiosauroid, and *Rhaeticosaurus mertensi* that was found either as an early-diverging pliosaurid or to lie outside the basal branching of Pliosauridae, Rhomaleosauridae, and Plesiosauroidea. The Werther plesiosaur (GZG.V.000092) although scored for only 15 characters out of 270 (5.55%), has been consistently inferred as a pliosaurid in all MPTs though its precise placement could not be found due to the fragmentary nature of the material. In contrast, all analyses using the implied weighting inferred a monophyletic Pliosauridae (including *Thalassiodracon* that was found at the base of the clade but excluding *Rhaeticosaurus* that lay outside the basal branching of the three major plesiosaur clades). All these runs also reconstructed the Werther plesiosaur among pliosaurids though, again, the runs did not find a stable position for the OTU. As such, examination of the inferred trees and character state mapping does not enable to identify a certain character state combination in GZG.V.000092 that would be common for all results placing the specimen on the pliosaurid lineage, although, for example, the transformation of character 165 (the appearance of median ventral surface) from 0 (approximately flat or convex surface) to 1 (presence of a round midline keel) was among those that nested GZG.V.000092 close to some Late Jurassic thalassophoneans. It also occasionally connected the OTU with *Anguanax zignoi*.

## DISCUSSION AND CONCLUSIONS

We describe previously unreported occurrences of Pliensbachian plesiosaurs from two sites located in North Rhine-Westphalia, Germany. One of the new records, originating

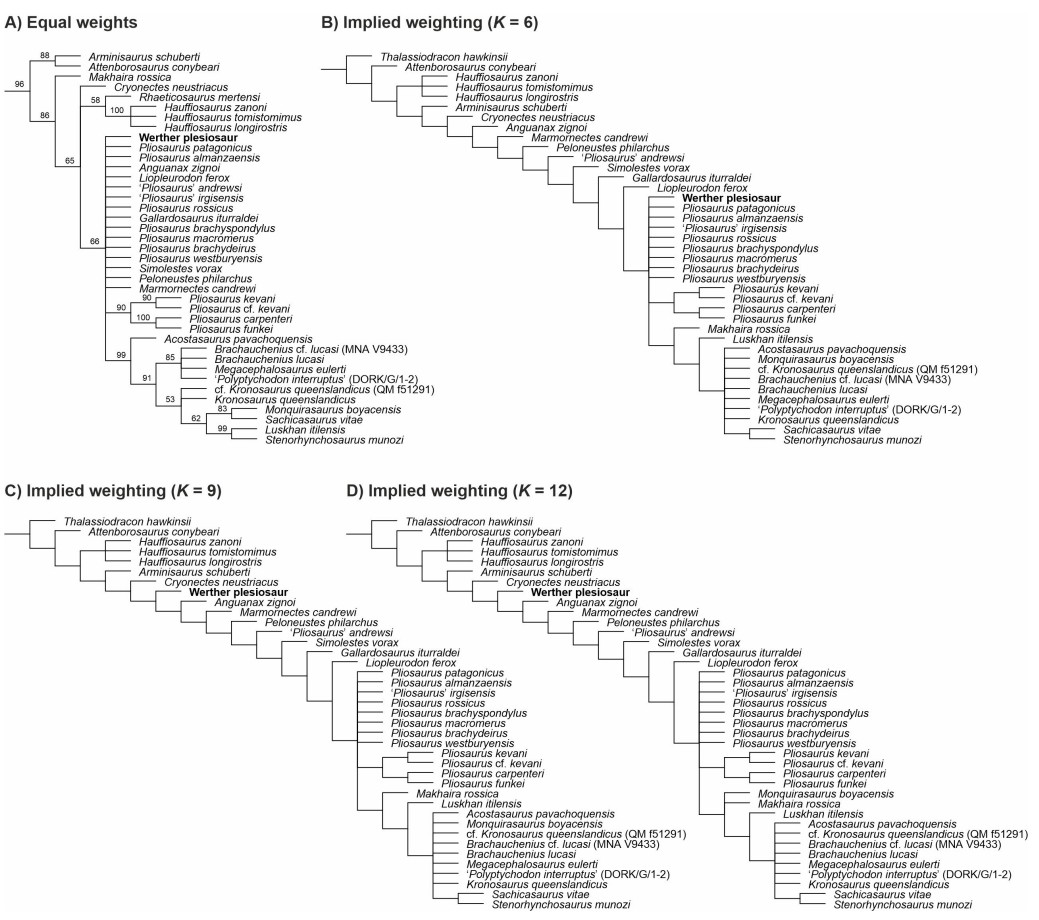

**Figure 7  The phylogenetic placement of the Werther plesiosaur (GZG.V.000092) shown on the pliosaurid segment of Plesiosauria.** (A) Reduced majority rule consensus tree reconstructed through parsimony analysis using equal weights (numbers at nodes indicate the percentage of the most parsimonious trees that found the nodes) and reduced strict consensus trees reconstructed through weighted parsimony analyses with *K* set to (B) 6, (C) 9, and (D) 12. The full topologies resulting from particular parsimony analyses, the Bremer support values for particular nodes inferred through the analysis using equal weights, and the results of Symmetric Resampling are provided in Supplemental Information 3.

from the *Uptonia jamesoni* Zone at Werther, represents the only reliably identified early Pliensbachian pliosaurid described to date. It is geographically and stratigraphically proximate to the late Pliensbachian pliosaurid *Arminisaurus schuberti* with which it shares a number of characters, including dorsoventrally facing cervical zygapophyses that are about as wide as the centrum, a rounded neurocentral suture, the presence of a prominent semicircular lip that extends ventrally from the anterior articular surface, triangular-shaped ventral bulge on the anterior and posterior articular facet, and a pronounced rounded ventral midline keel (see *Sachs & Kear, 2018*). A key character of *Arminisaurus*, the presence of parazygapophyseal processes between the pre- and postzygapophyses (*Sachs & Kear, 2018*: Fig. 4G), is not preserved in the Werther specimen. Similarities in the cervical anatomy can also be observed in *Cryonectes*, the only other currently known Pliensbachian

pliosaurid, although these largely reflect the early phylogenetic stage of the two specimens (*e.g.*, presence of subcentral foramina and two co-joined ventrolaterally-located rib facets, presence of a rounded, ventrally convex neurocentral suture). However, *Cryonectes* slightly differs from the Werther plesiosaur in having longer cervical centra, and lacking keels and ventrally-projecting lips on their ventral surfaces (*Vincent, Bardet & Mattioli, 2013*).

Owing to the lack of Pliensbachian plesiosaurs in general, which is manifested by an incomplete knowledge of the anatomy of their cervical region, it is currently difficult to infer the phylogenetic affinities of the Werther individual. Nevertheless, its character state combination indicates that it represents an early-diverging pliosaurid that was likely very similar, or perhaps even closely related, to the type of *Arminisaurus schuberti*. Even though it is currently impossible to identify the Werther specimen more precisely, its recognition increases the number of pliosaurid occurrences in Early Jurassic European epeiric seas.

**Institutional abbreviations**

| | |
|---|---|
| **GZG** | University of Göttingen, Göttingen, Germany |
| **Namu** | Naturkunde-Museum Bielefeld, Bielefeld, Germany |

## ACKNOWLEDGEMENTS

We thank Alexander Gehler and Lina Leschner (both GZG) for providing access to the material under their care. We are further indebted to Academic Editor Dagmara Żyła (Museum of Nature Hamburg, Leibniz Institute for the Analysis of Biodiversity Change, Hamburg, Germany) for handling our manuscript, and Mark Evans (British Antarctic Survey, Cambridge, United Kingdom), Espen Knutsen (James Cook University, Townsville, Australia), and Rodrigo Otero (Universidad de Chile, Santiago, Chile) for their constructive reviews.

### Funding

The authors received no funding for this work. TNT is made available with the sponsorship of the Willi Hennig Society. The funders had no role in study design, data collection and analysis, decision to publish, or preparation of the manuscript.

### Grant Disclosures

The following grant information was disclosed by the authors:
Willi Hennig Society.

### Competing Interests

The authors declare there are no competing interests.

### Author Contributions

- Sven Sachs conceived and designed the experiments, performed the experiments, analyzed the data, prepared figures and/or tables, authored or reviewed drafts of the article, and approved the final draft.

- Jahn J. Hornung conceived and designed the experiments, performed the experiments, analyzed the data, prepared figures and/or tables, authored or reviewed drafts of the article, and approved the final draft.
- Daniel Madzia conceived and designed the experiments, performed the experiments, analyzed the data, prepared figures and/or tables, authored or reviewed drafts of the article, and approved the final draft.

## Data Availability

The character matrix necessary to replicate the phylogenetic analyses is available in the Supplementary File.

## Supplemental Information

Supplemental information for this article can be found online at http://dx.doi.org/10.7717/peerj.18408#supplemental-information.

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
