# Peer review of "Early-diverging plesiosaurs from the Pliensbachian (Lower Jurassic) of northwestern Germany"

_PeerJ, doi:10.7717/peerj.18408_

## Round 0.1 · original submission · Minor Revisions

Please, address all the comments and suggestions. There were some concerns regarding phylogenetic analysis, which should definitely be taken into account.

·

Basic reporting

The paper is generally well structured and written with all necessary information provided, with some minor exceptions which addition will improve the readability and reproducibility of the work – e.g. additional references needed in the introduction; brief historical background for each specimen to support assumptions made regarding their stratigraphic positions; Museum specimen numbers for the GZG specimen.

Further details are in the attached PDF.

Experimental design

The phylogenetic analysis using Maximum Parsimony is the standard for this area of study, and a comprehensive global dataset is used. However, I question the need for such a comprehensive analysis for the material described which only allows a scoring of ~5% of the total characters available. While useful for visualising and provide further support for assumed relationships between complex combinations of characters of taxa, with the small number of characters scored here I would be concerned that the phylogenetic analysis results present a circular argument than an independent interpretation (i.e. the characters scored are readily attributed to pliosaurids, therefore the taxon will group with other pliosaurids).

Further details are in the attached PDF.

Validity of the findings

The conclusion appears to be based on the combination of observations of both the new specimens described, although no arguments have been presented to justify the assumption that they belong to the same taxon. For their current conclusions to be valid, the authors either need to argue for why they assume the two specimens belong to the same taxon or for a pliosaurid affinity for the Namu specimen. Otherwise it is possible that the GZG specimen is attributable to a known taxon (A. schuberti) and the Namu specimen belongs to an unknown taxon which may not be a pliosaurid.

Further details are in the attached PDF.

Additional comments

The new material constitutes a really important new datapoint for plesiosaurs and pliosaurs, more specifically, during this time period, and provide a starting-point for further work on this significant time in the origin of these groups. As such, I think with some minor adjustments to the manuscript, the work will be an important source for ongoing and future studies in this area.

I have provided more detailed comments in the PDF of the manuscript.

Reviewer 2 ·

Basic reporting

-While I'm not an english native speaker, the redaction and general language was easy to read and without ackward sentences.
-The literature used is up-to-date.
-The general structure is fine, but it can be improved. A general chapter of Material (maybe as Material and Methods) is missing. This is relevant because one of the specimens (GZG unnumbered) seems to be an historical material, but no background about it is provided in the current text. It would be interesting to know why the specimen lacks stratigraphic information.
On the Systematic Paleontology, the Remarks are barely a couple of lines. Maybe these could be incorporated together with the Description and Comparisons under the same sub-title.

Experimental design

-The current version of the manuscript does not provide strong evidence for referring the GZG (unnumbered) to the Pliosauridae. While an extensive phylogenetic analysis is provided, these lack indication of the characters and states supporting the inclusion of the GZG within the Pliosauridae. The description also lacks mention of those characters supporting this referral. This information is pivotal, because the stronger goal of the manuscript is the presence of an eventual pliosaurid different from Arminisaurus, in the Lower Jurassic of Germany. Otherwise, Lower Jurassic indeterminate plesiosaurians are known in other parts of the world, and therefore, the interest of the current contribution mostly resides in the validity of the GZG as a Pliosauridae non Arminisaurus.

On the other hand, while the phylogenetic analysis is extensive, the available material is much fragmentary, allowing a very scarce scoring percentage (16 characters for a total of 270; less than 6% of the total). With this, the results could be much artifactual, because most of the skeleton is actually, unknown. Said that, my recommendation is to focus on the available characters that could support a referral to the Pliosauridae, instead of a large phylogenetic analysis for a much fragmentary specimen (that could indeed, be reduced).

Validity of the findings

Please see my comments above. The referral to Pliosauridae needs to be strongly supported.

Additional comments

Aside of the previous comments, the current results of this manuscript are the presence of a (eventual) indeterminate Pliosauridae, and an indeterminate Plesiosauria in the Lower Jurassic of Germany. While the first could be eventually different from the previous known coeval forms, the much fragmentary condition of the material precludes any further comparison. Considering this, even if the manuscript review provides a strong support for the GZG specimen, I think the relevance of the finds are of limited interest, mostly local. None of the studied specimens provide new taxa or novel morphotypes.

Annotated reviews are not available for download in order to protect the identity of reviewers who chose to remain anonymous.

·

Basic reporting

The use of English in the article is clear, technically accurate and unambiguous except for one or two phrases that could benefit from rephrasing. The referencing is sufficient on the whole, although I have made a couple of suggestions for additional references. However, there are multiple examples (19) of references cited in the body of the text not listed in the references, which need to added.
The structure of the article is appropriate for a short descriptive paper such as this. The figures are valuable, useful and appropriate and are of sufficient resolution. I have made a couple of suggestions for figure 4; it would be useful if the ventral-most point of the neurocentral suture could be indicated, and check the anterior/posterior views in parts K and O. Two abbreviations (‘ld’ and ‘fs’)need to added to the caption for Figure 4.
The data provided allowed me to replicate the results using TNT, although please can you check the scoring of a couple of characters where TNT automatically converted this to range. Specifically, this was character 175 for OTUs Brancasaurus brancai and Arminisaurus schuberti where both were scored [0 2]. When I opened the file in TNT it converted these scores to 0-2, or states 0, 1 and 2.

Experimental design

The aims and design of the study are suitable for this descriptive type of study, with newly recognised fossil material being subject to anatomical description along with an assessment of its geological context and phylogenetic relationships. Although it could be argued that the material described is too fragmentary or meagre to warrant a dedicated descriptive paper, the study makes the case that the Pliensbachian represents a knowledge gap in the plesiosaur record and that every occurrence should be investigated.

Validity of the findings

The statement that the Werther material is the “only reliably identified early Pliensbachian pliosaurid” (line 252) is supported by its inclusion in the Pliosauridae in all MPTs of the phylogenetic analysis. However, the OTU is only 5.6% complete, being scored for 15 characters out of 270. It has only one of the characters recovered as a pliosaurid synapomorphy in the EW analysis, character 172(0), which is widely distributed among plesiosaurians. Given that it is debateable that it does show state 0 rather than the more ventrally placed convex neurocentral suture (state 3), I think there should be more discussion of the evidence supporting its pliosaurid identity.

---

## Round 0.2 · accepted · Accept

I confirm that the authors answered all reviewers' questions and commented on their suggestions. I am happy with the current form of the manuscript and in my opinion, it is ready for publication.